# AgNWs@TiO₂ and AgNPs@TiO₂ Double-Layer Photoanode Film Improving Light Capture and Application under Low Illumination †

**Jung-Chuan Chou** [1],*[ID], **Yu-Che Lin** [1], **Chih-Hsien Lai** [1], **Po-Yu Kuo** [1][ID], **Yu-Hsun Nien** [2], **Ruei-Hong Syu** [1], **Zhen-Rong Yong** [2] and **Yi-Ting Wu** [2]

[1] Graduate School of Electronic Engineering, National Yunlin University of Science and Technology, Douliou 64002, Taiwan; M10813072@yuntech.edu.tw (Y.-C.L.); chlai@yuntech.edu.tw (C.-H.L.); kuopy@yuntech.edu.tw (P.-Y.K.); M10913076@yuntech.edu.tw (R.-H.S.)

[2] Graduate School of Chemical and Materials Engineering, National Yunlin University of Science and Technology, Douliou 64002, Taiwan; nienyh@yuntech.edu.tw (Y.-H.N.); M10815027@yuntech.edu.tw (Z.-R.Y.); M10915028@yuntech.edu.tw (Y.-T.W.)

\* Correspondence: choujc@yuntech.edu.tw

† This paper is an extended version of our paper published in Lin, Y.-C.; Chou, J.-C.; Lai, C.-H.; Kuo, P.-Y.; Nien, Y.-H.; Chang, J.-X. Hu, G.M.; Yong, Z.-R. Effects of AgNWs@TiO₂ and AgNPs@TiO₂ Double-layer Structure on Electron Transfer and Light Capture of DSSCs. In Proceedings of the International Electron Devices & Materials Symposium 2020, Taoyuan, Taiwan, 15−16 October 2020.

**Abstract:** In this article, silver nanowires (AgNWs) were prepared and introduced into the double-layer photoanode of dye-sensitized solar cells (DSSCs). Silver nanowires with a diameter of about 50–60 nm and a length of 1–2 mm were prepared by the polyol method. The power conversion efficiency of the double-layer photoanode DSSC made of AgNWs@TiO₂ and AgNPs@TiO₂ composite materials is 6.38%. Compared with the unmodified DSSC, the composite double-layer photoanode combined with AgNWs and AgNPs increased the efficiency of DSSC by 58.7%. This increased efficiency was mainly due to the localized surface plasmon resonance effect caused by AgNPs and AgNWs. The increased light collection was caused by the plasma effect of AgNPs, and it increased the short-circuit photocurrent density ($J_{SC}$). The conductive properties of AgNWs improved interface charge transfer and delay charge recombination. The effect of a low light environment on DSSC efficiency was also investigated, and the best photovoltaic conversion efficiency under an irradiance of 10 mW/cm² was found to be 8.78%.

**Keywords:** AgNWs; AgNPs; low illumination; dye-sensitized solar cells; photovoltaic conversion efficiency

## 1. Introduction

The conventional dye-sensitized solar cell (DSSC) photoanode film is usually composed of a porous mesoporous layer of TiO₂ nanoparticles [1,2]. In order to develop low-cost [3], stable, and high-performance solar cells, various components of a DSSC have been extensively studied. Research on these key components included developing better photoanodes [4–6], synthesizing efficient sensitizers, and developing new electrolytes [7–9]. A TiO₂ nanomaterial photoanode composed of and modified by noble metals can increase the specific surface area and increase the adsorption of dyes; manufacturing a photoanode with appropriate structural modification can also significantly improve the efficiency of a DSSC. In addition, since the TiO₂ nanomaterial photoanode composed of and modified by noble metals can provide more efficient electron transfer, it has attracted widespread attention in the DSSC field in recent years [10].

The localized surface plasmon resonance (LSPR) effect caused by gold or silver precious metal nanoparticles has been reported to enhance the light trapping in DSSC applications [3,10,11]. It is believed that different metal nanostructures based on the different

combinations of $TiO_2$ nanomaterials will have an impact on the whole efficiency of DSSCs. These can implement light trapping or scattering technology to improve the performance of the photoanode. For example, this can be done by incorporating silver nanoparticles, nanotubes, or nanospheres into the $TiO_2$ photoanode. However, the doped of the noble metal will cause the film adhesion and quality to decrease. In order to improve this problem, we studied the preparation of a double-layer photoanode film, and different layers were modified by silver nanoparticles (AgNPs) and silver nanowires (AgNWs) [11], respectively. According to the experimental results, the photovoltaic performance of the modified DSSC has been improved. The AgNPs@$TiO_2$ layer causes the LSPR effect, thereby enhancing the interface charge transfer to improve the light trapping of the photoanode [12,13]. The AgNWs@$TiO_2$ charge conduction layer has a high-speed conduction channel to delay the charge recombination process, and this improves the electrical conductivity of the photoanode.

## 2. Materials and Methods

### 2.1. Reagents and Materials

The photoanode and counter electrode substrate of the DSSC was used fluorine-doped tin oxide glass (FTO glass, ~7 $\Omega$/cm$^2$, Sigma-Aldrich, St. Louis, MO, USA). For the AgNWs@$TiO_2$/AgNPs@$TiO_2$ photoanode film, P25 titanium dioxide powders ($TiO_2$, UniRegion Bio-Tech, Taoyuan, Taiwan), titanium dioxide nanoparticles (anatase 80%: rutile 20%, UniRegion Bio-Tech, Taoyuan, Taiwan), ethanol ($C_2H_5OH$ 99.98%, Katayama Chemical, Osaka, Japan), ethylene glycol (EG, $HOCH_2$–$CH_2OH$, Katayama Chemical, Osaka, Japan), silver nanoparticle (AgNPs 10 nm, Conjutek, New Taipei, Taiwan), Triton X-100 ($C_{14}H_{22}O(C_2H_4O)_n$, PRS Panreac, Barcelona, Spain), and acetylacetone (AcAc, Sigma-Aldrich, St. Louis, MO, USA) were used. A platinum target (Pt 99.99% purity, Ultimate Materials Technology Co., Ltd., Zhubei, Taiwan) was used to sputter the PT counter electrode. The DSSC assembly was assembled using ruthenium-535-bisTBA (N719, Solaronix, Aubonne, Switzerland) and an iodine-based electrolyte ($I_2$, MPN, TBP, LiI, DMPII, Sigma-Aldrich, St. Louis, MO, USA).

### 2.2. Preparation of Silver Nanowires by Polyol Method

Figure 1 shows a schematic illustration of the fabrication route for the AgNWs [14–16]. We synthesized the silver nanowires with the polyol method, and the diameter and length of the silver nanowire were controlled. First, we prepared an $AgNO_3$ precursor solution by mixing 0.45 g $AgNO_3$ and 5 mL ethylene glycol (EG). In order to prepare polymer protection solutions of different proportions, 4.16 g, 3.34 g, and 2.5 g polyvinylpyrrolidone (PVP) were mixed with 0.06 g NaCl, and 20 mL EG, respectively. Afterward, the polymer protection solution was heated to 150 °C. Then, we used the titration method to add the $AgNO_3$ seed solution at an injection speed of 5 mL/min. After the solution was reacted at 190 °C for 2 h, it was purified using a centrifuge. Finally, we used a freeze dryer to dry the AgNWs solution into a powder at −40 °C for future preparation into a film colloid.

### 2.3. Preparation of Film Colloid

The AgNWs@$TiO_2$ spin colloid consisted of 1.5 g P25 $TiO_2$ powder, 3 mL deionized water (D. I. water), 0.1 mL absolute alcohol, and different mass percentages of AgNWs (1 wt%, 2 wt%, 3 wt%). The composition of the AgNPs@$TiO_2$ doctor blade colloid was 2 g P25 $TiO_2$ powder, 0.4 mL absolute alcohol, and 1 wt% commercially available 10 nm silver nanoparticles. In addition, we also prepared pure $TiO_2$ colloidal colloids without precious metal doping. Each colloid was stirred for one day to ensure its uniformity.

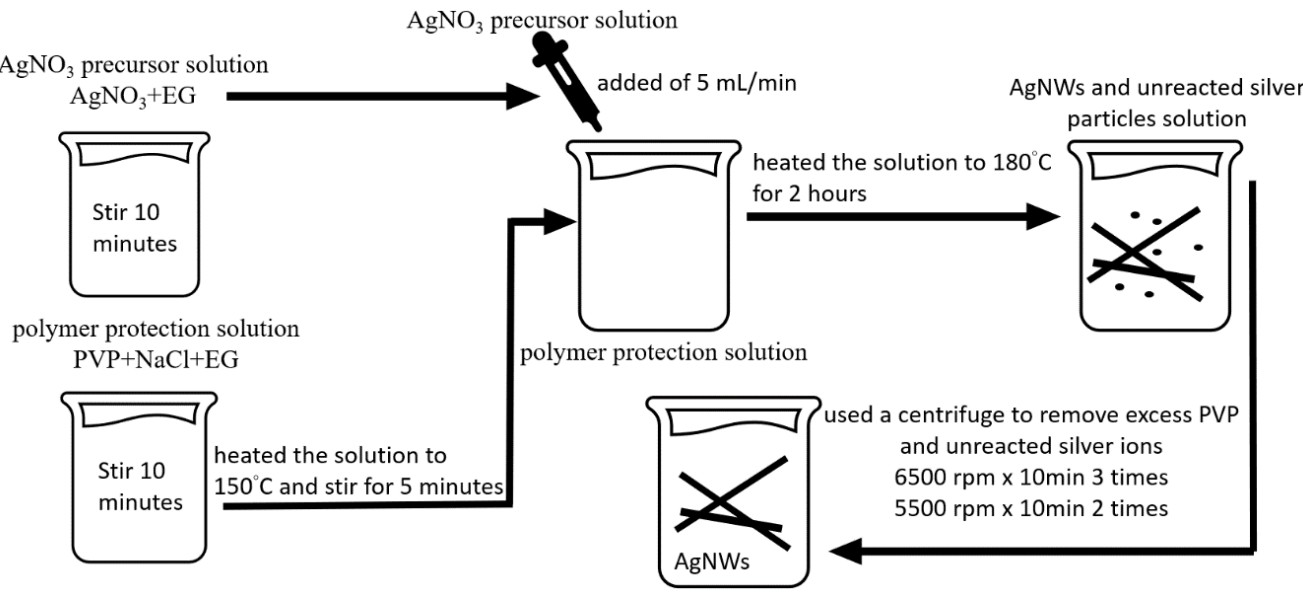

**Figure 1.** Schematic diagram of the manufacturing route of AgNWs prepared by polyol method.

### 2.4. Fabrication of DSSCs

We used FTO glass washed with acetone, ethanol, and deionized water as the substrate of the photoanode. Heat-resistant tape was used for setting the working area for the double-layer photoanodes, and the effective area of each photoanode was set to 0.25 cm$^2$. The AgNWs@TiO$_2$ layer was deposited on the FTO glass by the rotation method at an initial speed of 5000 rpm (10 s) and a final speed of 6500 rpm (12 s). Then, we used the doctor blade method to deposit the AgNPs@TiO$_2$ layer, and then we annealed the photoanode at 450 °C for half an hour. Then, the photoanode was soaked in 0.3 mM N719 dye for 24 h. In this study, we prepared five kinds of photoanode films including pure TiO$_2$ (DSSC 1), 1 to 3 wt% AgNWs@TiO$_2$/TiO$_2$ (DSSCs 2, 3, and 4), and 3 wt% AgNWs@TiO$_2$/1 wt% AgNPs@TiO$_2$ (DSSC 5), respectively.

### 2.5. Measurements and Characterization

The ultra-high-resolution thermal field emission scanning electron microscope (JSM-7610F Plus, FE-SEM, Tokyo, Japan) and high-resolution transmission electron microscope (JEM-2100 Plus, HR-TEM, Tokyo, Japan) were used to investigate the morphology of AgNWs and photoanodes film. The photocurrent–voltage curves of different photoanodes were conducted under an Xe lamp solar simulator (MFS-PV-Basic-HMT, New Taipei, Taiwan) and Keithley 2400 source meter; each sample was measured under 100 mW/cm$^2$ illuminance. We used a potentiostat/galvanometer (France BioLogic SP-150) in a dark room to measure the electrochemical impedance spectroscopy (EIS). The Nyquist plot was measured over the frequency range of 1 to 50 MHz with a potential perturbation of 10 mV.

## 3. Results

### 3.1. Morphology of AgNWs and Photoanode Thin Film

Figure 2 shows the FE-SEM image cross-section view for the photoanode based on FTO glass. The surface of the FTO glass was covered with a conductive layer of AgNWs with a thickness of approximately 4.13 μm and a light scattering layer of AgNPs with a thickness of 9.11 μm. Each photoanode was limited to an optimal thickness of about 13 μm [17]. In this study, AgNWs were manufactured by the reported polyol method [18,19]. The polyol method is a simple method to synthesize AgNWs by growing reduced Ag ions. During the synthesis process, the increased temperature will affect the length and diameter of AgNWs. The different mass percentage of silver nitrate and PVP will affect the

effectiveness of the capping agent [20]. Figure 3a–c shows the SEM images of the AgNWs synthesis with silver nitrate and PVP in molar ratios of 1:5, 1:4, and 1:3, respectively. In Figure 3a,b, we can observe that there were many silver nanoparticles. In our experiment, we prepared more completed and finer AgNWs with the best reaction ratio (1:3) as shown in Figure 3c. Furthermore, we used TEM to further analyze the surface morphology of the AgNWs. According to Figure 4a, it is clear that the AgNWs have a length of 1–2 mm. In Figure 4b, it can be found that the silver nanowire has a diameter of about 45–55 nm and a pentagonal structure. In our experiment in which the ratio of $AgNO_3$ and PVP was 1:3, the prepared result has a better characterization and uniform wire diameter results. Then, we used it as a modifier of DSSC photoanodes.

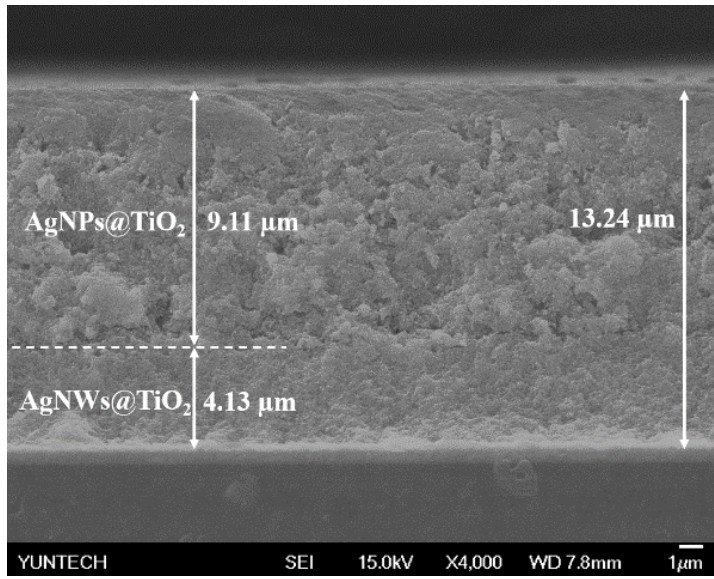

**Figure 2.** Field emission scanning electron microscope (FE-SEM) image of AgNWs@TiO$_2$/AgNPs@TiO$_2$ photoanode cross-section view.

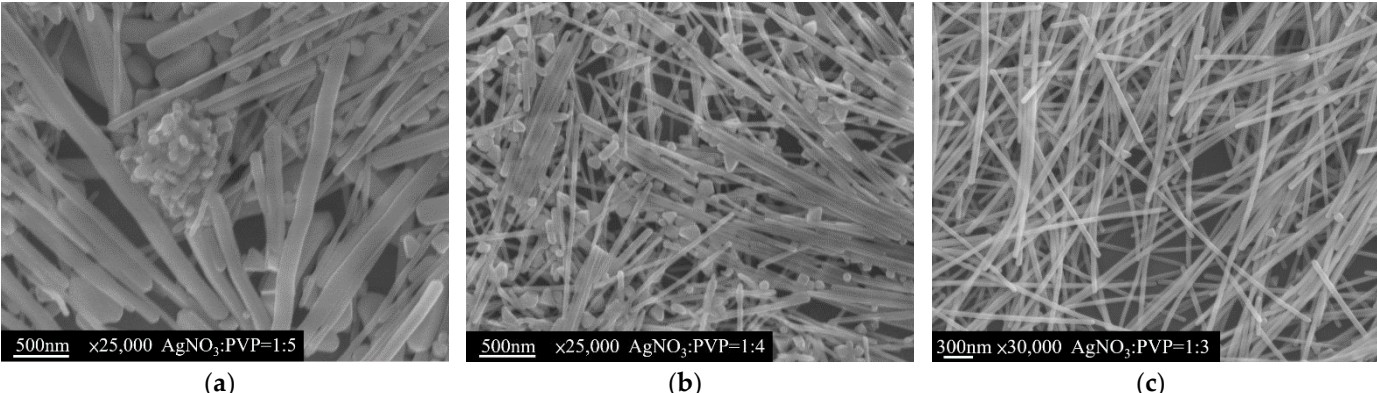

**Figure 3.** FE-SEM images of different reaction ratios AgNWs. AgNO$_3$:PVP = (**a**) 1:5, (**b**) 1:4, (**c**) 1:3.

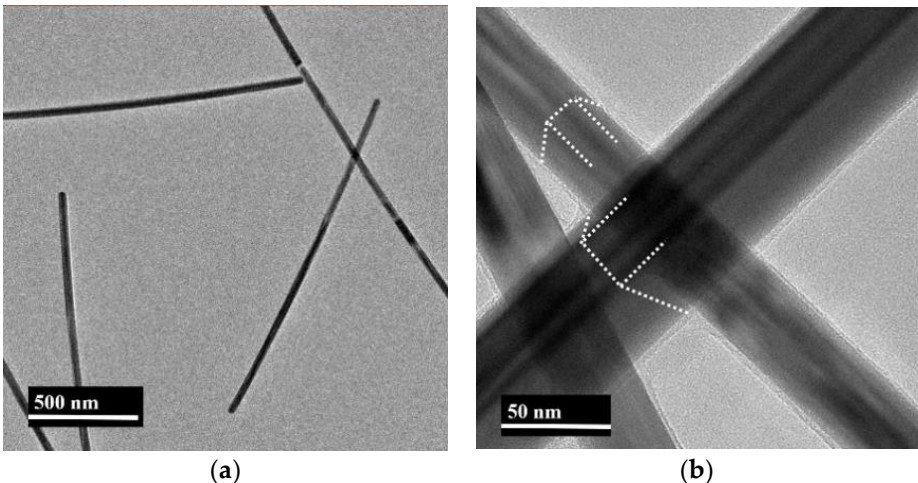

(**a**)  (**b**)

**Figure 4.** Transmission electron microscope (TEM) images of AgNWs at (**a**) 10k times and (**b**) 100k times magnifications.

### 3.2. Photovoltaic Performance of DSSCs

Figure 5 shows the photovoltaic performances of the DSSCs with the different photoanodes. Table 1 listed the corresponding photovoltaic parameters of different photoanodes. From the double-layer photoanodes made by different composite materials shown in Figure 5 and Table 1, the performance of the DSSCs improved the short-circuit photocurrent density ($J_{SC}$). The increased $J_{SC}$ and efficiency when AgNWs@TiO$_2$ was used to replace the charge transport layer were due to the conductive properties of AgNWs. The open-circuit voltage ($V_{OC}$) error of each sample was very small at about ±0.01 V. For the DSSCs with different weight percentages of AgNWs, their $J_{SC}$ reached to 8.26, 9.18, 10.04, 11.15, and 15.02 (mA/cm$^2$), respectively. The material properties of AgNWs provided excellent electronic conductivity of the photoanode, and the unique LSPR effect of nanometals significantly improved the light capture, which was the reason for the increased $J_{SC}$.

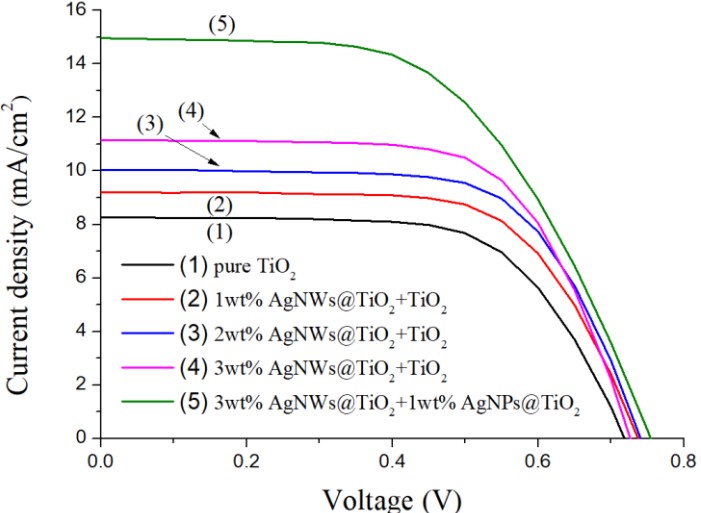

**Figure 5.** J-V characteristics of different photoanodes.

**Table 1.** Photovoltaic parameters of different photoanodes.

| Photoanode | $V_{OC}$ (V) | $J_{SC}$ (mA/cm$^2$) | Fill Factor (%) | η (%) |
|---|---|---|---|---|
| DSSC 1 (pure TiO$_2$) | $0.719 \pm 0.00$ | $8.26 \pm 0.11$ | $64.65 \pm 0.73$ | $4.02 \pm 0.15$ |
| DSSC 2 (1 wt% AgNWs@TiO$_2$ + TiO$_2$) | $0.74 \pm 0.01$ | $9.18 \pm 0.23$ | $61.99 \pm 0.68$ | $4.47 \pm 0.25$ |
| DSSC 3 (2 wt% AgNWs@TiO$_2$ + TiO$_2$) | $0.74 \pm 0.01$ | $10.04 \pm 0.20$ | $66.31 \pm 0.59$ | $4.93 \pm 0.16$ |
| DSSC 4 (3 wt% AgNWs@TiO$_2$ + TiO$_2$) | $0.73 \pm 0.01$ | $11.15 \pm 0.14$ | $65.49 \pm 0.79$ | $5.31 \pm 0.06$ |
| DSSC 5 (3 wt% AgNWs@TiO$_2$ + 1 wt% AgNPs@TiO$_2$) | $0.75 \pm 0.01$ | $15.02 \pm 0.23$ | $56.72 \pm 0.68$ | $6.38 \pm 0.25$ |
| 3 wt% AgNWs + P25 [21] | 0.75 | 12.01 | 0.59 | 5.31 |
| 4 wt% AgNWs + P25 [21] | 0.76 | 10.44 | 0.59 | 4.70 |
| Ag-TiO$_2$ [22] | 0.78 | 11.10 | 0.65 | 5.62 |

Huang et al. [21] reported that the improved efficiency of DSSC made by AgNWs@TiO$_2$ is due to the decrease of the interface charge transfer resistance and the ion diffusion resistance in the electrolyte. It is worth noting that with the increase of nanometal, one-dimensional nanostructures (AgNWs) are overloaded, and the host (TiO$_2$) is not able to maintain the electron path. The breakage and fragmentation of the film can reduce the quality of the film, as indicated by the gradual decrease in the fill factor (FF). These results indicate that there are limitations in increasing the amount of doping. To prove that the LSPR effect caused the light absorption to increase, we also measured quantum efficiency. The incident photon-to-electron conversion efficiency (IPCE) curves of the DSSCs are shown in Figure 6. Under normal circumstances, it can be determined that nanostructures with a diameter of less than 90 nm are very strong light absorbers and nanostructures with a diameter greater than 90 nm help scattering [23]. It can be seen from the TEM image in Figure 4b that the average size of AgNWs synthesized by the polyol method is about 45–55 nm in diameter, indicating that these AgNWs will strongly absorb visible light through the LSPR effect [12,13]. In the IPCE curves, it was found that AgNWs enhanced the light capture around wavelength region 550 nm. Compared with the DSSC 1 made of pure TiO$_2$, the J$_{SC}$ and efficiency of the DSSC 4 are more prominent. Compared to the TiO$_2$ photoanode, the IPCE value increased by 23%.

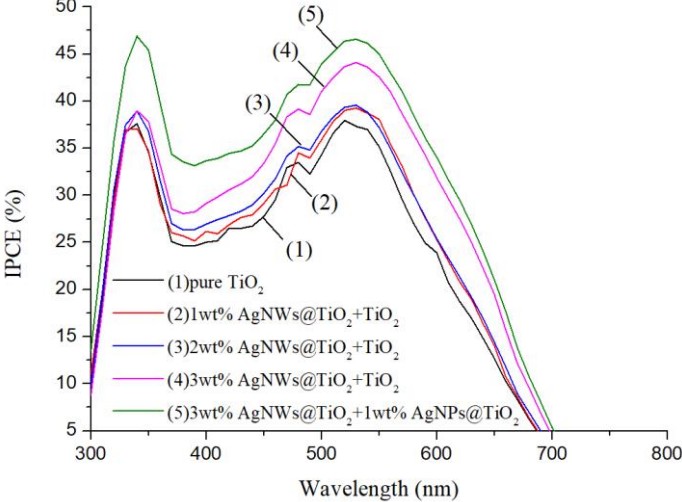

**Figure 6.** Incident photon-to-electron conversion efficiency (IPCE) curves of the dye-sensitized solar cell (DSSCs) with containing different photoanodes.

In order to avoid the increase in defect density, DSSC 5 was prepared by adding silver nanoparticles to the scattering layer. As mentioned in several studies [22,24,25], too much modifier will affect the integrity of the photoanode film. According to Selvapriya et al. [22], relative to the surrounding medium, the local electric field enhancement around AgNPs is the main reason for better $J_{SC}$. In a specific excitation wavelength range (500 to 600 nm), the interaction between the dipole oscillation of free electrons and the N719 dye molecule can improve the photon collection ability to enhance device efficiency. In our experiment, the photocurrent density of DSSC 5 is increased to 15.02 mA/cm², and the photovoltaic efficiency is increased to 6.38%.

### 3.3. Electrochemical Impedance Spectroscopy of DSSC

To understand the interface electron conduction process, different electrochemical impedance analysis circuits and Nyquist plots of DSSCs are shown in Figure 7 [24]. Table 2 lists the charge transfer resistance ($R_1$, $R_2$), the imaginary capacitance ($C_1$, $C_2$) of the platinum counter electrode/electrolyte interface, and the photoanode/electrolyte interface; $R_S$ stands for contact resistance.

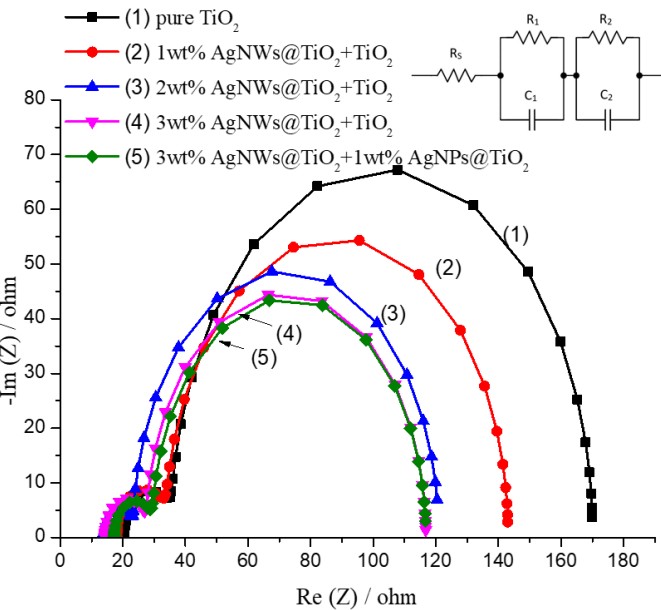

**Figure 7.** Equivalent circuit and Nyquist plots of the DSSCs with containing different photoanodes.

**Table 2.** Electrochemical impedances spectroscopy (EIS) for DSSCs with different photoanodes.

| Photoanode | $R_S$ (Ω) | $C_1$ (μF) | $R_1$ (Ω) | $C_2$ (mF) | $R_2$ (Ω) | $\tau_n$ (ms) |
|---|---|---|---|---|---|---|
| DSSC 1 (pure TiO₂) | 20.46 | 18.2 | 15.05 | 0.29 | 134.6 | 39.03 |
| DSSC 2 (1 wt% AgNWs@TiO₂ + TiO₂) | 18.06 | 19.2 | 15.67 | 0.34 | 109.3 | 37.16 |
| DSSC 3 (2 wt% AgNWs@TiO₂ + TiO₂) | 13.89 | 21.2 | 13.41 | 0.48 | 97.1 | 46.60 |
| DSSC 4 (3 wt% AgNWs@TiO₂ + TiO₂) | 14.25 | 19.1 | 13.21 | 0.54 | 89.3 | 48.22 |
| DSSC 5 (3 wt% AgNWs@TiO₂ + 1 wt% AgNPs@TiO₂) | 17.23 | 18.1 | 12.12 | 0.54 | 87.4 | 47.19 |

The composite material of AgNWs@TiO₂ could promote charge separation and improved the electron-transfer dynamics. Compared with the $R_2$ resistance value of the TiO₂ DSSCs, the introduction of AgNWs improved the electronic recombination of the photoanode/electrolyte interface [25]. When the AgNWs included in the electronic conductive layer increased, the electron transfer resistance of the TiO₂/electrolyte interface

($R_2$) gradually decreased from 134.6 Ω to 87.4 Ω. It can be seen that $R_2$ decreased with the increase of the content of AgNWs content, which means that the appropriate content of AgNWs enabled better electronic contact between $TiO_2$ and FTO glass. The decreased resistance of the $TiO_2$/dye/electrolyte interface ($R_2$) could be due to the AgNWs promoting the interface electron transfer in the photoanodes. The AgNWs with a good Fermi level (EF~0.4 V) can be used as a good electron acceptor [26]. Therefore, the electron flow from the conduction band of $TiO_2$ (ECB = −0.5 V vs. NHE) into the lower Fermi level of AgNWs is feasible [27]. Figure 8 schematically shows the entire recombination reaction that occurs in the AgNWs@$TiO_2$ photoanodes. In the AgNWs@$TiO_2$ based DSSCs, electrons excited by the N719 sensitizer were injected into $TiO_2$ after light irradiation. The electron was transferred from the $TiO_2$ conduction band to the AgNWs, thereby creating a charge balance in the device. The free-electron transfer of AgNWs caused the metal nanocrystals to be in a nonthermal equilibrium state. The Schottky barrier existing at the Ag and $TiO_2$ interface makes the electrons in the conduction band of $TiO_2$ neither return to the dye nor return to the electrolyte. In other words, photogenerated electrons tend to be injected from $TiO_2$ into AgNWs to form high-speed conduction channels. Since electrons injected into $TiO_2$ will be quickly transferred to AgNWs, this can suppress the reverse electron conduction. This situation attributed to the energy level of AgNWs being lower than that of the $TiO_2$. The reduction of charge recombination at the photoanode/electrolyte interface helps to reduce the dark current and at the same time increases the $J_{SC}$, thereby increasing the overall efficiency. The electron lifetime ($\tau_n$) can be calculated by Equation (1) [28].

$$\tau_n = R_2 \times C_2 \tag{1}$$

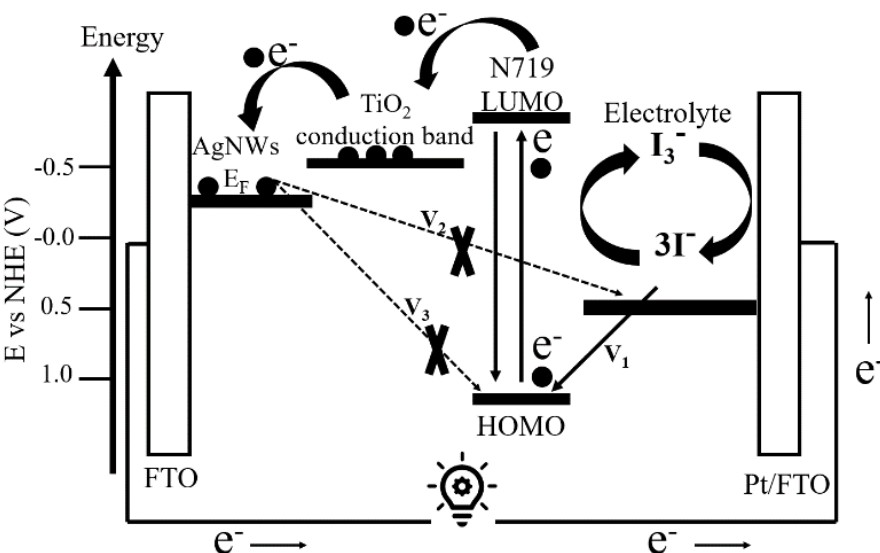

**Figure 8.** Schematic illustration of the interfacial charge transfer recombination process that occurs with the AgNWs@$TiO_2$ based DSSCs. $V_1$ represents the redox reaction between the oxidation N719 molecules and the iodine electrolyte. $V_2$ represents the recombination reaction between injected electrons and redox electrolytes. $V_3$ represents the recombination reaction between injected electrons and oxidized N719 molecules.

According to Table 2, the photoanodes film with different weight percentages AgNWs prolonged the electron lifetime. The $\tau_n$ was increased from 39.03 to 48.22 ms and is inversely proportional to $R_2$. Longer $\tau_n$ indicated that DSSCs devices can use photogenerated electrons more effectively. The electron lifetime of photoanodes film covered with 1 wt% AgNPs@$TiO_2$ layer was shortened to 47.19 ms, which may be due to the reduction of redox species in the porous semiconductor.

### 3.4. Performance of DSSC under Different Light Intensities

Figure 9 and Table 3 showed the J-V curves and performance parameters of DSSC under different illuminations. With reducing light intensity, the photoelectric conversion efficiency of the DSSC modified with 3 wt% AgNWs were increased from 5.31% to 6.17%. The PCE of DSSCs modified with 3 wt% AgNW and 1 wt% AgNP increased to 7.42%. In a low-illuminance environment, the incident photons are reduced, which results in less sensitizer dye being excited. The reduction in $J_{SC}$ and $V_{OC}$ is attributed to fewer photogenerated electrons, which also means lower output power. As a result of the silver nanoparticles on the photoanode surface, the charge recombination was suppressed, and this ensured more effective utilization of visible light radiation. Although $V_{OC}$ and $J_{SC}$ decrease with light intensity, the utilization rate of photon energy was gradually increased, and the filling factor was also gradually increased. According to Huang et al. [29], the incident light power was too low under the illumination of 10 mW/cm$^2$ so that the electrons were intercepted by the oxidized electrolyte. Nien et al. [30] showed that under low illumination, DSSCs can reduce the recombination reaction to enhance PCE, and the LSPR effect of AgNPs made more effective use of visible light radiation. In our work, under the low illumination of 10 mW/cm$^2$, the Schottky barrier interacting with AgNWs and TiO$_2$ inhibited the recombination of charge and electrolyte, and the PCE increased to 8.78%.

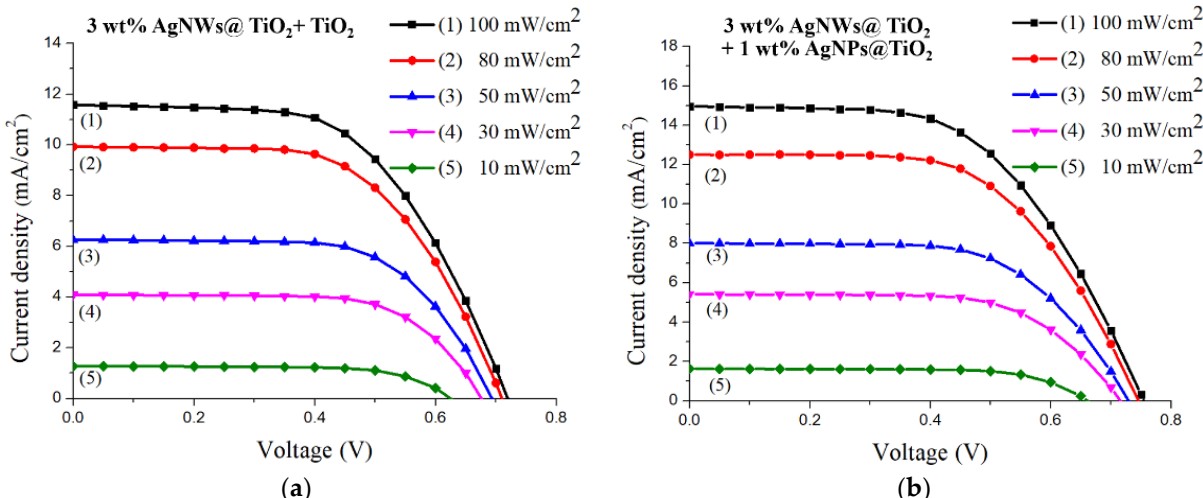

**Figure 9.** J-V curves of DSSCs under different light intensities. (**a**) 3 wt% AgNWs@TiO$_2$/TiO$_2$ and (**b**) 3 wt% AgNWs@TiO$_2$/1 wt% AgNPs@TiO$_2$.

**Table 3.** Photovoltaic parameters of different photoanodes under different light intensities.

| Photoanode | Intensity (mW/cm$^2$) | $V_{OC}$ (V) | $J_{SC}$ (mA/cm$^2$) | Fill Factor (%) | η (%) |
|---|---|---|---|---|---|
| 3 wt% AgNWs@TiO$_2$ + TiO$_2$ | 100 | 0.73 ± 0.02 | 11.15 ± 0.14 | 65.49 ± 0.79 | 5.31 ± 0.06 |
| | 80 | 0.72 ± 0.01 | 10.80 ± 0.15 | 56.64 ± 0.56 | 5.35 ± 0.05 |
| | 50 | 0.71 ± 0.00 | 7.21 ± 0.21 | 62.73 ± 0.19 | 5.84 ± 0.03 |
| | 30 | 0.71 ± 0.01 | 4.67 ± 0.17 | 65.10 ± 0.96 | 6.17 ± 0.05 |
| | 10 | 0.65 ± 0.00 | 1.52 ± 0.09 | 70.29 ± 0.67 | 6.49 ± 0.09 |
| 3 wt% AgNWs@TiO$_2$ + 1 wt% AgNPs@TiO$_2$ | 100 | 0.75 ± 0.01 | 15.02 ± 0.23 | 56.72 ± 0.68 | 6.38 ± 0.25 |
| | 80 | 0.74 ± 0.01 | 12.38 ± 0.12 | 58.81 ± 0.49 | 6.40 ± 0.16 |
| | 50 | 0.73 ± 0.01 | 8.12 ± 0.09 | 62.54 ± 0.11 | 6.83 ± 0.20 |
| | 30 | 0.72 ± 0.00 | 5.31 ± 0.12 | 65.10 ± 0.26 | 7.42 ± 0.11 |
| | 10 | 0.62 ± 0.01 | 1.72 ± 0.04 | 70.59 ± 0.37 | 8.78 ± 0.05 |
| Counter electrode of CoSe$_2$/CoSeO$_3$-NP [29] | 100 | 0.81 | 15.88 | 71 | 9.27 |
| | 50 | 0.78 | 8.11 | 72 | 9.31 |
| | 10 | 0.72 | 1.82 | 72 | 9.41 |
| TiO$_2$/Ag NF [30] | 100 | 0.73 ± 0.01 | 10.05 ± 0.09 | 69.92 ± 0.51 | 5.13 ± 0.16 |
| | 80 | 0.72 ± 0.02 | 8.29 ± 0.12 | 70.75 ± 0.47 | 6.40 ± 0.14 |
| | 50 | 0.71 ± 0.01 | 5.60 ± 0.14 | 71.84 ± 0.48 | 5.71 ± 0.12 |
| | 30 | 0.70 ± 0.01 | 3.67 ± 0.13 | 72.66 ± 0.45 | 6.23 ± 0.16 |
| | 10 | 0.69 ± 0.02 | 1.08 ± 0.13 | 70.83 ± 0.46 | 5.31 ± 0.13 |

## 4. Conclusions

We reported an advanced double-layer structure photoanode film, which achieved an efficiency of 6.38% under standard illuminance. At a light intensity of 10 mW/cm$^2$, the power conversion efficiency was 8.78%. This double-layer device structure was doped, with which AgNWs were prepared by a polyol method and AgNPs, respectively. Based on this device structure, the LSPR effect was employed by mesoporous TiO$_2$ films which were doped with nanometal. The enhanced light trapping will lead to more photogenerated electrons, thereby boosting the J$_{SC}$. In addition, AgNWs provided a good conduction path for electrons and thus weaken the internal interface impedance R$_2$ of DSSCs. This structure not only improved the indoor photovoltaic conversion efficiency but also accelerated the DSSCs to become a new type of indoor light collection equipment.

**Author Contributions:** J.-C.C. proposed the subject of the study; Y.-C.L. wrote a first draft of the manuscript, performed J-V and EIS measurements; C.-H.L. provided consultation on optical measurement; P.-Y.K. contributed to literature updates and manuscript revisions; Y.-H.N. contributed to AgNWs synthesis and material analysis; R.-H.S. and Z.-R.Y. performed the SEM and TEM studies of the selected samples; Y.-T.W. performed the measurement of DSSC samples. All authors have read and agreed to the published version of the manuscript.

**Funding:** This research has been supported by the Ministry of Science and Technology, Taiwan, R.O.C., under the contract MOST 109-2221-E-224-013.

**Informed Consent Statement:** Not applicable.

**Data Availability Statement:** The data presented in this study are available on request from the corresponding author.

**Conflicts of Interest:** The authors declare no conflict of interest with the research performed and reported here.

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
