# Peer review of "AgNWs@TiO2 and AgNPs@TiO2 Double-Layer Photoanode Film Improving Light Capture and Application under Low Illumination†"

_chemosensors, doi:10.3390/chemosensors9020036_

Round 1

Reviewer 1 Report

The paper presented develops a very relevant approach for the development of efficient DSSCs. The approach proposed develops previous experiences on localized surface plamon resonance and its role on the improvement of charge transfer and delay charge recombination.

For a better understanding of the principles involved the authors should develop a better background presentation of the charge transfer and recombination principles relevant for the research.

It is also important to check table 2 data (C1 and C2 values are missing) followed by a detailed explanation of the experimental Nyquist plots for a clearer understanding of the achieved results.   

Reviewer 2 Report

In this work, the authors synthesized the silver nanowires (AgNWs) and used them as the double layer photoanode of DSSC. Compared with the unmodified DSSC, the composite double-layer photoanode combined with AgNWs and AgNPs increased the efficiency of DSSC. This title and studied contents have some certain interests in the field of chemistry, energy, and materials. But there are still some problems that need to be solved by the authors to improve the quality of the manuscript. The specific questions are as follows:

(1)authors should check the statement in manuscript ‘This paper is an extended version of our paper published in Effects of AgNWs @ TiO2 and AgNPs @ TiO2 Double-layer 15 Structure on Electron Transfer and Light Capture of DSSCs. 2020.’

*Authors should confirm that the neither current manuscript nor any parts of its content are currently under consideration or published in another journal.

(2) Line 88-89: the unit “ml” should be revised as “mL”. Please check and revise the similar errors in the manuscript.

(3) Figure 1: please check the writing standard of units, such as min, ml, hours, etc.

(4) What is the effect of the thickness of AgNWs on the efficiency of DSSC? The authors did not explain this point. Please add discussion.

(5) in the manuscript, the name of the fig and table should be ‘Fig and Table’. Word initial is capitalized. At the same time, the writing forms of figures and tables are not uniform in the discussion. Please revise them.

(6) Although the experimental results in this work are well, the authors should add some explanation for the internal mechanism of DSSC performance improvement by theoretical simulation.

(7) In Table 3, the PCE of the DSSCs are increased along with the decreasing light intensity. In the real-life, is the lower the light intensity, the better for DSSC to realize photoelectric conversion?

(8) Figure 7: the Nyquist plots of the DSSCs look like a broken line. Is this due to the parameter setting of the experimental instrument, or is it related to the structure of DSSCs? please add discussion.

(9) In section 3.2, performances of DSSC under different light intensities. Table 3. Photovoltaic parameters of different photoanodes under different light intensities. Usually, the experimental tests are under AM1.5G. sunlight with 100mW/cm2. Before doing experimental tests, generally standard si solar cells are utilized to make an adjust. Does the author has used a standard si solar cells to make an adjustment as a reference? The author should make an explanation.

(10) some English grammar errors should be revised. For example, ‘line 38-42 of page 1, To properly modified photoanode films can increase the specific……’; ‘…TiO2 nanomaterial photoanode composed modified by noble metals’, composed can be deleted; sentence in line 50-52 of page 2 ‘How-50 ever, the increase in the amount of noble metal doping is not that simple, such as the de-51 crease in adhesion to the substrate, the occurrence of film cracking, and the difficulty of 52 bonding with TiO2.’ Should be rewritten because of complex and not scientific expression.; the sentence in line 131-132 ‘The silver nanowire images with molar ratios of 1:5 fig. 3(a) and 1:4 fig. 3(b) were observed many silver nanoparticles.’ Should be rewritten; in line 192-193, ‘…gradually decreases, from 134.6 Ω decreases to 87.4 Ω.’ should be revised to be ‘gradually decreases, from 134.6 Ω to 87.4 Ω.’; the sentence in line 245-247 ‘As shown in Table 3, compared with reference [30] when the light intensity of 30 mW/cm2 was down to 10 mW/cm2, due to the interaction of AgNWs and AgNPs, PCE increases by 18%.’ be rewritten, and English structure and expression are confusing and difficult to understand.

(11) in conclusion, for the sentence in line 229-230 ‘Discussed the influence of AgNWs and AgNPs on the photoanode, and measured the DSSCs under low illumination.’, The sentence lacks subject so that the sentence should be rewritten. it is suggested that the authors should check and correct the English grammar and syntax carefully. At the same time, for conclusions, instead of simply describing the experimental results, the authors should summarize the highlights of this work and look forward to future improvements.

Reviewer 3 Report

The authors have synthesized Ag NWs with TiO2 as photoanode, and by using it with Ag NP/TiO2 as double layers in DSSCs, they found an enhanced performance, which is believed to be related to plasmonic effects. However, the results and discussion regarding the solar cell performance are not clear, and more results are needed. Moreover, the novelty of this work and the significance of the findings in this work needs to be improved. The language is also required to check and revised. Overall, the reviewer suggests to reject it for the current version, and can be reconsidered if the authors do a comprehensive revision. The comments are followed:

  1. In the introduction, the authors claim that the Ag NWs can induce plasmonic effects and contribute a higher Jsc in DSSCs. However, there is no evidence or results to show this effects. The reviewer suggests to give more discussion on this point.
  2. From SEM images, it shows the Ag NWs have a long diameter about 1mm, however, NWs are not seen from the cross sectional image in Figure 2. What is the reason for that? 
  3. High temperature is used to anneal TiO2 and Ag NW film, is the Ag NW stable enough under such high temperature without causing any damage?
  4. The idea of such double layer structure is to induce plasmonic effects, but the authors mentioned that Ag NW can influence the charge transport properties, which sounds confusing. The key component in photoanode is TiO2 and dye molecules, the excited electrons inject into TiO2 directly through dyes and transport through TiO2 nanostructures. In principle, there is no contribution from Ag to the electron transport process. Please explain this.
  5. Can the Ag NW affects the dye loading? If dyes can be adsorbed on Ag NWs, can they contribute more Jsc? How about the transparency of TiO2/Ag NWs comared to TiO2? Does Ag NW decrease the light harvesting?
  6. More results and clear discussion are needed to explain the changes in photovoltaic parameters for double layer photoanode?

Round 2

Reviewer 2 Report

The manuscript entitled " AgNWs @ TiO2 and AgNPs @ TiO2 Double-layer Photoanode Film Improving Light Capture and Application Under Low Il-3 lumination". The author has reported the modification of photoelectrode to improve the efficiency of solar cells. After the first round, there are also some problems the author should pay attention, which are listed as follows:

1) in the introduction, the sentence ‘Gold or silver precious metal nanoparticles can through localized surface plasmon resonance (LSPR) strong scattering effect to more easily excite the electron transition of dye molecules’ should be rewritten for clear expression.

2) In section of 2.2, preparation of silver nanowares by polyol method, page 2, line 74, and the first time of ‘EG’ appeared should mark its in whole name; all the similar atmosphere the author should be paid attention to.

3) the sentences for the statement of molar ratios are not clear in lines 116-118. Please check the sentence ‘Fig. 3(a) and Fig. 3(b) are the AgNWs images with molar ratios of 1:5.’, because you state in the manuscript that ‘Figure 3 (a-c) showed the SEM 116 images of AgNWs synthesis with silver nitrate and PVP in molar ratios of 1:5, 1:4, and 1:3, 117 respectively. Fig. 3(a) and Fig. 3(b) are the AgNWs images with molar ratios of 1:5.’; if are two pictures all corresponding to the same molar ratios of 1:5 as you metioned? It is suggested to add the label of molar ratios into the Figure 3 (a-c).

4) sentence in line of 138-139 is ‘The DSSCs used with different weight percentage AgNWs, JSC reached are 8.26, 9.18, 10.04, 11.15, and 15.02 (mA/cm2), respectively.’ should be revised as ‘For the DSSCs with different weight percentage AgNWs, their JSC reached 8.26, …., respectively’. So some English expression errors should be checked for the whole manuscript.

5) Figure 5, J-V characteristics of different photoanodes should have connection with Figure 6, IPCE curves of the DSSCs with containing different photoanodes. While there are some different parts. In figure 5, the five curves include (1) pure Tio2 (2) 1wt% AgNW@TiO2+TiO2 (3) 2wt% AgNW@TiO2+TiO2 (4) 3wt% AgNW@TiO2+TiO2 (5) 3wt% AgNW@TiO2+3wt%AgNPs@TiO2.   While there are some different in figure 6, which includes four parts, (1) pure TiO2 (2) 2wt% AgNW@TiO2+TiO2 (3)  3wt% AgNW@TiO2+TiO2 (4) 3wt% AgNW@TiO2+1wt%AgNPs@TiO2. The author should make an explanation why in figure 6 that the curve of 1wt% AgNW@TiO2+TiO2 has been lost. And why one composition has changed (in figure 5, 3wt% AgNW@TiO2+3wt%AgNPs@TiO2; while in figure 6, 3wt% AgNW@TiO2+1wt%AgNPs@TiO2).

6) The Schottky barrier existing at the Ag and TiO2 interface makes the electrons in the conduction band of TiO2 neither return to the dye nor to the electrolyte. Can the author explain clearly why does the existence of the Schottky barrier existing at the Ag and TiO2 interface can make the electrons neither return to the dye nor to the electrolyte. Especially for inhibit the electrons back to the electrolyte.

7) It is suggested that the line of 176-188 should be rewritten because the physical process in the doped metal DSSCS is not clear. The author should add the schematic of charge transfer processes in TiO2 /AgNWs (and AgNPs)/dye/electrolyte interface or schematic structure (or energy lever for the plasmonic solar cells, PSCS) in supporting materials, which is very helpful to explain the microscopic mechanism of charge transfer for PSCs.

Reviewer 3 Report

The authors have tried to answered most of the questions, and the manuscript has been improved. However, the role and working mechanism of Ag NW in photoanode has still not been demonstrated clearly, which would be worth to study more in future work. Therefore, the reviewer suggests to accept it after the authors carefully checking the language and details in manuscript (format, references etc).
